# Screening the Impact of Surfactants and Reaction Conditions on the De-Inkability of Different Printing Ink Systems for Plastic Packaging

**DOI:** 10.3390/polym15092220

**Published:** 2023-05-08

**Authors:** Jinyang Guo, Cong Luo, Christian Wittkowski, Ingo Fehr, Zhikai Chong, Magdalena Kitzberger, Ayah Alassali, Xuezhi Zhao, Ralf Leineweber, Yujun Feng, Kerstin Kuchta

**Affiliations:** 1Circular Resource Engineering and Management (CREM), Hamburg University of Technology (TUHH), Blohm Str. 15, 21079 Hamburg, Germany; 2Siegwerk Druckfarben AG_Co.KGaA, Alfred-Keller-Str. 55, 53721 Siegburg, Germany; 3Polymer Research Institute, State Key Laboratory of Polymer Materials Engineering, Sichuan University, Chengdu 610065, China

**Keywords:** flexible plastic packaging, de-inking with surfactant, printing ink

## Abstract

One of the major applications (40% in Europe) of plastic is packaging, which is often printed to display required information and to deliver an attractive aesthetic for marketing purposes. However, printing ink can cause contamination in the mechanical recycling process. To mitigate this issue, the use of surfactants in an alkaline washing process, known as de-inking, has been employed to remove printing ink and improve the quality of recyclates. Despite the existence of this technology, there are currently no data linking the de-inking efficiency with typical printing ink compositions. Additionally, it is necessary to investigate the de-inking process under the process parameters of existing recycling plants, including temperature, NaOH concentration, and retention time. This study aims to evaluate the performance of commonly used printing inks with different compositions under various washing scenarios for plastic recycling in conjunction with different de-inking detergents containing surfactants or mixtures of surfactants. The results indicate that the pigments applied to the ink have no significant effect on the de-inking process, except for carbon black (PBk 7). Nitrocellulose (NC) binder systems exhibit high de-inkability (over 95%) under the condition of 55 °C and 1 wt.% NaOH. However, crosslinked binder systems can impede the de-inking effect, whether used as a binder system or as an overprint varnish (OPV). The de-inking process requires heating to 55 °C with 1 wt.% NaOH to achieve a substantial effect. Based on the findings in this work, breaking the Van der Waals forces, hydrogen bonds, and covalent bonds between the printing ink and plastic film is an essential step to achieve the de-inking effect. Further research is needed to understand the interaction between surfactants and printing inks, enabling the development of de-inkable printing inks and high-performance surfactants that allow for de-inking with less energy consumption. The surfactant and NaOH have a synergistic effect in cleaning the printing ink. NaOH provides a negative surface charge for the adsorption of the cationic head of the surfactant and can hydrolyze the covalent bonds at higher concentrations (>2 wt.%).

## 1. Introduction

According to recent statistics, the global consumption of plastic exceeded 400 Mio. Mg in 2019, while nearly the same amount was generated as plastic waste [1]. In Europe, the annual plastic consumption was reported to be 50 Mio. Mg in 2020, with 40% being used for packaging applications [2]. Low-density polyethylene (LDPE) accounted for 8.8 Mio. Mg/a and most of it was used in the form of a flexible film [3]. Other polymers (e.g., high-density polyethylene (HDPE), polyethylene terephthalate (PET), polypropylene (PP)) can also be converted into flexible packaging such as pouches, bags, sleeves, and labels [2,4,5]. 

### 1.1. Flexible Plastic Waste Management

Flexible plastic waste (i.e., films and labels) can be categorized into two types: post-industrial and post-consumer sourced [6]. The post-industrial flexible packaging waste includes production scraps (i.e., from packaging production). Post-industrial plastic waste is normally considered a high-quality input stream for mechanical recycling due to its homogeneity and lower purity content [7]. However, some limitations also exist in the recycling of post-industrial flexible packaging waste. For instance, the printed production scrap for food packaging cannot be applied again for food packaging production since printing inks contain many substances that are not known to the recycler and could degrade or react during extrusion to yield an even more significant number of potential substances. According to European food safety regulations, unknown substances have to be treated as genotoxic [8].

Post-consumer flexible plastic waste can be further categorized as (1) from commercial operation—this material stream has a similar homogeneous quality to the post-industrial waste—and (2) from municipal solid waste (MSW) or segregate waste collection from the household. In Europe, the effective recycling rate of post-consumer flexible packaging waste was reported to be 17% [9]. 

The flexible plastic waste is separated normally in a sorting plant by air classification in wind shifters. Due to their comparatively large surface and low weight, flexibles move with the air upwards against gravity and can be separated from rigid components. Further sorting steps can include hand sorting or automatic sorting with near-infrared (NIR) spectroscopy [10,11,12]. Currently, the focus in recycling flexible plastic packaging is LDPE (e.g., “fraction 310”, a term coined by the German dual-system “Grüner Punkt”) [13]. Therefore, multilayer packaging and films made from other types of polymers (i.e., PP) are often not targeted and removed in this process [14]. The multilayer fraction is commonly reported as a problem in recycling and is often used for energy recovery in incineration [12,15]. Recent studies reported the feasibility of sorting multilayer from single-layer fractions. However, its industrial application is in an early stage [16,17].

Following the sorting step, the flexible plastic waste is subjected to the recycling process, which is composed of shredding, density separation, washing, drying, and extrusion [18]. Several technologies are applied in the recycling process to improve the quality of the recyclate, including hot washing with NaOH, extrusion–degassing, melt filtration, and hot-air deodorization. However, these technologies are reported to be only implemented in a limited number of advanced recycling plants [13,19].

Currently, flexible packaging is often down-recycled into low-value products [9,13,20,21]. Due to contamination, flexible plastic waste packaging (FPWP) is converted for other applications into a material with a higher thickness (i.e., traffic cones and garden furniture) with injection molding [22]. Another typical market for recycled LDPE is the thick black film used for waste bags or construction applications, where carbon black or blue masterbatches are added to achieve a uniform color [3].

Contamination is a major obstacle in the recycling of plastic packaging. This is not limited to product residues but also includes the polymer and non-polymer elements of the packaging itself [14,23]. The polymer contamination originates mainly from imperfect sorting processes. For instance, PP is a common source of contamination in the LDPE plastic stream. Achieving complete separation of PP and LDPE flexibles during the sorting process is challenging with current compressed air sorting technology. Non-targeted materials (i.e., PP labels) can end up unintentionally sorted into the LDPE film stream when they are conveyed close to each other at the air nozzle area. In the recycling plant, PE and PP cannot be separated via the conventional sink–swim process with water since both polymers have a similar density of <1 g/cm^3^ [24]. Non-polymer contamination includes additives, glue, processing agents, printing inks, and their degradation and side reaction products [25,26,27]. Lastly, contamination from the recycling process can also deteriorate the quality of the recyclates, e.g., surfactants or solvents [28].

### 1.2. Printing Ink as a Source of Contamination

Flexible packaging is a major contributor to the packaging waste stream, and the printing ink used on the surfaces of flexible packaging is a significant source of contamination [29,30]. Printing ink and coatings are necessary components for packaging, as it provides product information and enhances the visual appeal of the product for marketing purposes [30]. The printing ink of flexible packaging material contains mainly pigments that provide the color, a binder that keeps the pigment particle in place and provides the adhesion on the packaging surface, and additives that provide additional characteristics of the packaging, such as the coefficient of friction, anti-block, scuff resistance, and more [31,32]. Some printing ink formulations are multilayer structures, with a varnish layer without pigments printed on top of the ink layer that contains pigments, known as overprinting varnish (OPV) [33]. Before the printing process, the printing ink is suspended in solvents called carriers. The printing inks can be therefore categorized into solvent-based (organic solutions) and water-based (water with alcohol) inks [32]. These types of inks and coatings require a drying process to evaporate the solvents and water. There are also inks that “dry” without any evaporation of water or solvents in chemical reactions. UV and electron beam curing inks are typical technologies in this class that are widely used on various sleeves and label applications but also, e.g., in specific applications such as aluminum lids [34]. Due to the low surface tension of the semi-crystallized polyolefins, surface treatment with corona discharge is a standard process for improving the adhesion of the ink on plastic surfaces [35]. The corona discharge introduces light oxidation, forming hydroxy, carbonyl, and carboxyl groups on the plastic surface and anchors with the isocyanide (only present in two-component ink formulations), hydroxy, and carboxyl groups in the binder resin [36,37,38]. The solvents evaporate during the printing process. Therefore, the solvent is of little relevance in the recycling stage. After the printing process, only the binder, pigment, and additives remain in the printing ink system attached to the printed surface.

The common chemicals applied in printing inks for plastic packaging are shown in Table 1.

The presence of printing ink on flexible plastic packaging creates several issues during the mechanical recycling process. First, the mixing of the different pigments results in a dark or grayish color after the extrusion process, which lowers the market value of the recyclates [30]. Secondly, recyclates with printing ink mixed inside were reported to have lower tensile strength properties [39,40]. Finally, some ink components are not stable under the extrusion temperature of the mechanical recycling process (150–240 °C) [29,41,42,43]. Nitrocellulose (NC) is the most commonly used binder resin for flexible plastic packaging printing inks due to its low price and high gloss [44]. However, NC is vulnerable to thermal degradation starting at 160–185 °C, producing CO_2_ and NO_2_, resulting in a brownish color, porosity, and odor in the recyclates [45,46,47,48,49]. Additionally, some organic pigments were also reported to be thermally unstable under mechanical recycling processes, such as pigment PR 146 and pigment PY 13 [50].

### 1.3. Surfactant Cleaning in the Recycling Process

The removal of printing ink residue from the surface of flexible plastic packaging can be achieved through detergency, mechanical, and chemical cleaning processes [51]. Examples of mechanical cleaning processes are the roll-to-roll de-inking machines offered by Gamma Meccanica from Italy and Polymont BV from the Netherlands [52,53]. They are mostly used on post-industrial film and apply high friction with brushes (the latter applies organic solvents in the roll cleaning machine). The majority of flexible plastic waste is ground into flakes during the mechanical recycling process. The cleaning and friction occur in friction washers, centrifuges, and other fast-running mechanical components [10,18]. Chemical removal involves organic solvents to dissolve the printing ink on the film surface [54]. Although cleaning printed flexible plastic waste with organic solvents is possible, no industrial-scale application has been reported since the organic solvents for the de-inking application are highly volatile, have a low flash point, and are more expensive [55,56,57]. Thus, the water-based detergency cleaning process is a potentially more suitable solution for removing printing ink residue from LDPE film surfaces.

The detergency cleaning process generally comprises two sub-processes: the removal of dirt (also known as “soil”) from the substrate and the suspension of the soil in the cleaning solution (also known as the “bath”) [58]. For the case of plastic de-inking, the cleaning process was reported to involve the following steps [59]:Adsorption of the surfactant on plastic surfaces;Removal of the printing ink (soil);Detachment of the mixture of ink particles and surfactant from the surface;Stabilization of the detached ink particles.

Soil removal is the most crucial step in de-inking as it breaks the mechanical (intersurface diffusion) and chemical adhesion (hydrogen bonds, covalent bonds, and Van der Waals force) between the plastic surface and the printing ink [60,61,62]. Four different removal mechanisms were summarized as (1) roll-up, (2) spontaneous emulsification, (3) soil softening, and (4) encapsulation [63]. Since all de-inking experiments (Table 2) were carried out above the critical micelle concentration (CMC) of the surfactant, the dominant mechanism for plastic de-inking should be the solubilization of the binder in the micelle, known as solubilization or encapsulation. 

The earliest research on cleaning plastic films with surfactants was carried out by Gecol et al. using nonionic surfactants, including ethoxylated nonylphenol, ethoxylated alcohol, and ethoxylated amine, which were reported to be effective in a solution with a pH over ten and heating under 55 °C [64]. A series of studies were conducted to test different surfactants (anionic, cationic, nonionic, and zwitterionic) on water-based and solvent-based ink systems with different colors [39,64,65]. These researchers reported that water-based ink could be dissolved without a surfactant at a high pH value (11–13). Quandary ammonium cations present the best de-inking performance, especially cetrimonium bromide (CTAB) [65].

An overview of the publications and industrial applications of plastic detergency de-inking processes are shown in Table 2.

To the best of the authors’ knowledge, there is no systematic research on the de-inkability of different ink components, including binders, pigments, and over-printing varnish, with respect to various factors, including temperatures, pH values, surfactants, and retention times. In previous studies, the printing ink systems were categorized by the type of carrier, as “solvent-based” and “water-based” inks. As the carrier is evaporated in the printing process, no relevance of the de-inkability and the binder system was reported. Furthermore, there was no information about the pigment investigated. Only the different colors were reported. 

Based on the review of all the previous studies, as well as relevant commercialized processes, there are knowledge gaps regarding the de-inkability of plastic packaging printing inks of different compositions. Therefore, we aim to accomplish the following in this study:To evaluate the de-inkability of common printing inks characterized by their components with existing surfactants for plastic cleaning in the recycling process, reported in the literature;To provide an open-source method for de-inkability evaluation;To derive process parameters to integrate the de-inking process into current plastic recycling plants and future “green-field” plants.

## 2. Materials and Methods

The samples (Table 3) are films with common surface printing ink systems available on the market, provided by Siegwerk Druckfarben AG & Co.KGaA (Siegburg, Germany). The sample matrix is agreed upon by Siegwerk, Sun Chemical, and Flint to represent the current printing ink market for flexible plastic packaging. One exception is Sample 12, which is a single-layer PU ink without OPV. This construction does not exist on the market and is only produced for research purposes.

The sample matrix was designed to evaluate the de-inkability of printing ink for plastic packaging with different ink components. Samples 1–5 have a controlled variable of pigment. Sample groups 2, 11, 13, 14 and 12, 13 were designed to investigate the influence of binder systems without OPV on the de-inkability of NC and PU binder systems, respectively. Samples 4, 7, 8, 9, and 10 were selected to investigate the de-inkability of crosslinked binder systems. It should be noted that Sample 12, the PU binder without OPV, is not found in reality. Without OPV or any crosslinking, the PU is too soft and sticky to survive the utilization phase of printed plastic packaging.

This study uses five different detergents, which were reported in different studies as cleaning agents in plastic recycling (Table 4). Sodium hydroxide was purchased from Carl Roth GmbH. As some cleaning agents are mixtures, the term “detergent” is used in this paper, and the term “surfactant” is used when the single component of the detergent is discussed.

Four different test conditions were applied in order to test the cleaning performance of the detergents. The first condition (A) simulates the currently existing recycling plants without the addition of NaOH (“cold washing”). The temperature was set at 40 °C, as this is the temperature applied in the washing process without heating for a state-of-the-art recycling plant. The temperature increase comes from mechanical friction from mixing, friction washing, and other aggregates. Condition C represents the “hot-washing” condition used in some recycling plants [72]. Therefore, condition B was set as the middle point of conditions A and C in terms of temperature and sodium hydroxide consumption to investigate the potential of reducing the energy consumption from condition C. Condition D was only used for samples that could not be de-inked under conditions A–C, reported by studies for separating multilayer flexible packaging [72]. An overview of the different testing conditions is presented in Table 5.

Each sample was tested with each surfactant and test condition in combination until satisfactory de-inking (95%) was achieved. The first step of the experiment was to prepare the solution. A 400 mL glass beaker was placed on a balance and tared. Using a Pasteur pipette, 0.5 g of surfactant was added to the beaker (for liquid surfactants). Next, 0.2, 0.4, or 1.0 g of NaOH was added based on the test condition. Finally, the beaker was filled with water to a total weight of 200 g. The beaker was then secured to a heating plate with a rubber belt. After the addition of the sample flakes to the beaker, the agitation speed was set to 800 rpm. The color change was measured using a colorimeter. After the set retention time, the sample flakes were separated from the liquid phase using a coarse sieve. The liquid phase was collected for flocculation. The experimental procedure is shown in Figure 1.

### 2.1. Evaluation of the De-Inking Effect

The de-inking performance was assessed by comparing the color difference before and after de-inking. Nine flakes were randomly selected from each sample, glued with a colorless glue onto DIN A4 white print paper and scanned in a scanner (EPSON ET-5880). The L-a-b values according to the CIELAB standard [73,74] were measured with a self-developed program with Python 3.10, and the program code is given in Appendix A. The color change of the samples before and after the de-inking was calculated with the same method applied in previous de-inking studies [40,64,65,67], shown in Equations (1) and (2).
(1)$${\mathrm{DE}}^{*}=\sqrt[{2}]{\Delta L^{2}+\Delta a^{2}+\Delta b^{2}}$$
where:

DE^∗^: color difference;

∆L = L_sample_ − L_blank_;

∆a = a_sample_ − a_blank_;

∆b = b_sample_ − b_blank_.
(2)$${\mathrm{DI}}_{i,j,k}=1-\frac{{\mathrm{DE}}_{i,j,k}^{*}}{{\mathrm{DE}*}_{i,\;\mathrm{original}}}$$
where: 

DI: de-inking rate (100 = complete de-inking, 0 = no de-inking);

i: sample number (Table 3);

j: detergent (I–VI, Table 4);

k: condition (A–D, Table 5).

### 2.2. Evaluation of Detergents Applied

To evaluate the hydrophilic–lipophylic balance (HLB) of the detergents, we used Davies’ method, shown in Equation (3). The HLB values of the different functional groups were taken from the same references [75,76].
(3)$$\mathrm{HLB}=7+\overset{m}{\underset{o=1}{\sum }}H_{o}-0.475n$$
where:

H_i_: value of the hydrophilic groups;

m: number of hydrophilic groups;

o: the hydrophilic groups;

n: number of hydrophobic groups.

### 2.3. Water Treatment

The collected liquid phase was stirred at 150 rpm while adding 1 *v/v*% flocculation agent (SEPAR Chemie AW+KW). The flocculate was removed via vacuum filtration with 15-micrometer porosity filter paper (Rotilabo Type 601A). Photometric analysis was carried out with a photometer with the wavelength scanning method from 900 to 300 nm.

## 3. Results

The results of the study are presented in three parts. The first part (Section 3.1, Section 3.2 and Section 3.3) focuses on the influence of various printing ink components on the de-inkability of the printing inks. This part details the specific properties of different ink components and their impact on the de-inking process. According to the visual observations during the experiment, a 95% de-inking rate can be regarded as complete de-inking. The second part (Section 3.4) focuses on the de-inking performance of different cleaning agents. The final part (Section 3.5) introduces the results of the flocculation efficiency after de-inking. This is a critical step in ensuring water reusability at recycling plants that carry out de-inking.

### 3.1. Influence of Different Pigments

Samples 1–5 were selected, with pigments being the only variable, thereby enabling an investigation into the impact of different pigments on the de-inking process. The experimental result is shown in Figure 2.

The color of the bars in Figure 2 is identical to the color of the pigments tested. From Figure 2, it can be concluded that all pigments, except Sample 4 (PBk 7), give results indicating comparable de-inking performance, with complete de-inking (>95%) taking place at condition B with all detergents, except III. Sample 4 showed the lowest de-inking effect (8.27% and 23.06%, respectively) under the test conditions III-A and B. This phenomenon is due to the black pigment (PBk 7) being mainly composed of highly hydrophobic carbon black, which cannot be dispersed by detergent III into the water phase, possibly due to its low HLB value (as discussed in Table 6) [77,78].

### 3.2. Influence of Different Binders without OPV

During the experiment on a single-layer printing ink system, two different de-inking behaviors were observed: solubilization and peeling. Solubilization occurred with the NC and PU binder systems. In this behavior, the printing inks were dispersed into the bath during the de-inking process, resulting in a cloudy bath with a homogeneous pigment color. On the other hand, peeling was observed when the PVB and UV acrylic binder system was inspected. In this behavior, the detached ink did not form a homogeneous suspension but instead peeled off in small flakes and behaved as sediment in the water phase.

For the NC binder, no de-inking effect was shown under condition A (Figure 3) except for detergent III (27.30%), and complete de-inking effects (95%) were obtained under condition B, except for detergent III (85%). This leads to the conclusion that the alkaline condition favors the de-inking effect, especially for cationic surfactants. The NC polymer carries a negative charge under alkaline conditions and favors the adsorption of cationic surfactants [79]. The de-inking mechanism of the PU binder should be similar to that in the NC system. However, it should be noted that this sample is de-inked at 40 °C, even without the use of NaOH, which renders it unsuitable for real-world applications, and it is only a sample especially created for this research. The de-inking of the single-layer binder system can be considered the standard solubilization process, shown in Figure 4a.

For the PVB and UV acrylate binder, where “peel off” effects were observed (Figure 4b,c), detergents had little contribution to the de-inking effect. No de-inking effect was shown under condition A. For condition B, the highest de-inking rate was measured with detergent VI (85.83%). Under condition C, all the detergents except III showed a complete de-inking effect (95%).

For the PVB binder, it is hypothesized that the de-inking mechanism is the hydrolysis of the chemical bonds between the binder and surface-treated plastic. Therefore, the high NaOH concentration contributed positively to the de-inking. In contrast, the low de-inking effect of III-A and B compared to the blank (I-A and B) supports this hypothesis. Due to the high hydrophobicity of surfactant W111, a protective layer is formed on the PVB surface, isolating it from contact with OH- groups.

For the UV-crosslinked binder system, de-inking effects were only found under condition C. As the monomer of the UV-crosslinked acrylate has a carboxyl group, it is also sensitive to the highly alkaline environment [39]. Additionally, its crosslinked structure further limits the diffusion of OH- groups into the binder, and this explains why the reaction only happened at 2 wt.% NaOH and 70 °C, as the temperature contributes positively to the diffusion. A similar protective effect was detected with detergents II, III, IV, and VI. The reason for the result of V-C as an exception is due to its emulsion effect specifically towards acrylate, which slightly positively contributes to the cleaning effect (99.09% vs. 98.87%) [80].

### 3.3. The Influence of OPV

#### 3.3.1. NC Binder with Crosslinked NC OPV

Sample 5 is a simple NC binder system without OPV, used as a zero measurement. Sample 8 is a double-layer construction, and both layers are non-crosslinked NC. Sample 7 and Sample 9 have a similar double construction with ink and OPV as Sample 8. However, the OPV for Sample 7 is crosslinked nitrocellulose with 13 wt.% hardener (crosslinking agent), and Sample 9 is CAP, which is also a derivate of cellulose, with 31 wt.% hardener.

Almost no de-inking effect was obtained from the highly crosslinked Sample 9. It is worth mentioning that the ink content of the four samples with different OPVs was 2.03, 2.05, and 2.85 g/m^2^, respectively. The addition of the simple OPV without crosslinking had no effect on the de-inking rate with detergents II, IV, and V. From the difference between the 13 wt.% and 31 wt.% samples in their de-inking rates, it can be concluded that the OPV limits the de-inking rate, which is not due to the increased mass of binder resin but due to the crosslinked structure.

From Figure 5, it can be observed that the addition of OPV has a limited impact on the de-inking rate when using detergents II, IV, and V, provided that these detergents are capable of achieving complete de-inking. For example, under condition C, Sample 7 with 13 wt.% crosslinked NC-OPV can be de-inked. However, Sample 9 with 31 wt.% crosslinked CAP-OPV showed no de-inking effect with all the detergents, even under condition D (Figure 6).

It is worth mentioning that, compared to the single-layered printing ink without OPV, the bonding mechanisms of the crosslinked OPV were different. The printing ink binder without OPV bonded with the substrate by the Van der Waals force. Due to the reactivity of the crosslinking agent, it is bonded by covalent bonds to the layer (printing ink or plastic film). The detergency washing cannot break the covalent bonds [58]. However, the conditions applied in condition C (2 wt.% and 70 °C) can hydrolyze some functional groups (i.e., ester, amino) [81,82]. Therefore, a relatively high de-inking rate can be obtained under this condition with Sample 7. The reason for the zero de-inking rate for Sample 9 is that CAP is highly crosslinked, and the hydrolysis effect under condition D is only limited to a substantial effect for detergent washing.

The de-inking effects of the samples with overprint varnish (OPV) were investigated, and it was determined that the de-inking mechanism was not solely due to detergency washing (micelle solubilization). Instead, it appears that chemical reactions may be occurring to break the covalent bonds in the binder system or between the binder system and the plastic film, thus synergizing the de-inking process.

#### 3.3.2. PU Binder with Crosslinked PVB OPV

The de-inking effect of the PU binder with the crosslinked PVB OPV system is shown in Figure 7. The PU binder without OPV can be de-inked in pure water at 40 °C. However, such an ink structure is not found in any commercial product. With the mandatory, protective OPV, PU is unable to de-ink the binder.

For Sample 13, with a crosslinked PVB as the OPV, no de-inking effect was obtained in conditions A and B. Under condition C, a high de-inking effect was obtained with detergent IV (94.80%). The contrast in detergents II and IV in the de-inking effect in this sample is quite surprising since the practical contents of II and IV are similar (CTAC and CTAB), only with different counter-ions. It can be concluded that the other contents in detergent II might have a hindering effect on the adsorption process or the solubilization process, specifically on crosslinked PVB, as the de-inking effects were similar with non-crosslinked PVB binders. One possible explanation is the protective effect due to the hydrophobic nonionic surfactant in detergent II.

For all the NC samples with crosslinked overprint varnishes (OPV), the de-inking rates are lower than their counterparts without OPV or non-crosslinked OPV. Therefore, the application of OPV is unavoidable for PU binders and only allows de-inking under conditions C and D. The surfactant micelle has limited internal spaces for accommodating large molecules with crosslinked structures [83].

### 3.4. Performance of the De-Inking Detergents

The overall performance of all six detergents used is shown in Figure 8a–d.

It can be concluded that the single-layer nitrocellulose binder showed the best de-inkability already under condition B by detergent washing. For the PVB binder, condition C is the most suitable condition for de-inking. The UV acrylate binder can be washed off only under condition D. Under condition A, detergent III showed the highest de-inking effect on average. However, the de-inking effect of detergent A is generally low, especially not suitable for de-inking the PVB binder system. During the experiment, it was commonly found that with detergent III, the printing ink could be softened even under condition A. The softened ink could be removed by introducing mechanical friction (brushing). Possibly due to its low HLB value (Table 6), the micelle was still adsorbed on the LDPE film surface and could not be solubilized into water. Under condition B, as depicted in Figure 8, detergents II and VI showed the best de-inking effects under conditions B–D.

**Table 6 polymers-15-02220-t006:** HLB values of the tested detergents.

Detergent	HLB *
I	Not applicable
II	7.22
III	5.33
IV	7.375
V	8.6
VI	4.8

* Calculated as the weighted sum of all the components given by the safety data sheet (SDS) [84].

Under hot-washing conditions C, surfactant IV showed the best de-inking effect. This is due to its superior performance with both Samples 13 and 14, which have crosslinked structures and are difficult to de-ink under milder conditions.

For condition D, all the detergents showed little improvement compared to washing without a detergent (I, with 5 wt.% NaOH). Implementing an alkaline hot-washing process with a detergent in recycling plants can reduce energy consumption.

The hydrophilic–lipophilic balance of the detergents tested is listed in Table 6. Concerning the different parameters of the detergents, it can be concluded that a detergent with an HLB value between 7 and 9 is preferable (suitable as wetting and spreading agents) [85]. A surfactant with a low HLB value (III) might hinder the de-inking process. However, this type of surfactant has the potential for further development due to its ability to soften the ink. This gives the possibility of designing a de-inking process without extra heating and the addition of NaOH, which could enable the integration of de-inking with lower effort regarding the modification of current recycling plants.

The interaction between the polymer and surfactant can be investigated by measuring the interfacial surface tension (IFT) between the soil and the bath by spinning a drop tensiometer [86]. However, this method is not applicable in this context, as the printing inks are mixed with organic solutions or water-dissolved organic solutions, resulting in a miscible system. Additionally, the structure of the printing ink can be changed by crosslinking during the print process (Samples 7, 9, 10, 13, and 14). Therefore, contact angle measurement using an optical contact meter could be a promising, simple method for comparing the wetting behavior of surfactants on different printing ink systems and non-printed film surfaces [87,88,89].

### 3.5. Water Treatment by Flocculation

For the wastewater treatment of the de-inking water, the first focus was to investigate the influence of the pigments, since they can directly change the color of the water. All the pigments applied in this study are not water-soluble. Therefore, they are solubilized by the surfactant. Since pigments contain much smaller molecules than the binder system, they are more likely to be solubilized in the micelle. Figure 9 shows the change in the light absorption of the different pigments under complete de-inking in condition B-I. Before flocculation, all the samples showed a significant increase in light absorption. After flocculation and filtration, the light absorption of the washing water was still two to four times higher than that of the virgin bath prior to de-inking.

The bonding capacity of the different binder systems is shown in Figure 10, where the de-inking baths of different samples (2, 11, 12, and 14) with different binders but the same pigment (PR 146) were investigated. UV acrylate showed the best bonding effect on the pigments before and after flocculation. This is due to its crosslinked structure, which was “peeled off” instead of being solubilized by the detergents. Interestingly, the PVB binder with a similar “peel off” behavior showed a more significant color change before and after flocculation, which leads to the conclusion that a crosslinked structure favors the stabilization of the pigment in the binder system.

## 4. Discussion

### 4.1. Design for Recycling in Plastic Packaging Printing Ink

One of the most important quality features of printing plastic packaging inks is their resistance against chemicals, scratches, and abrasion [90]. However, this contradicts the target of ink removal in the recycling process. Therefore, it can be concluded that a recycling-friendly ink should primarily maintain the performance in the utilization phase of the packaging. Ideally, they can only be washed off when NaOH and a detergent are used in the washing process.

From the overall results of the experiment, it can be concluded that nitrocellulose (NC) binder-based printing ink systems can be considered recycling-friendly if detergent-based alkaline hot washing is introduced in the recycling process. Even if crosslinked OPV is needed, it can still be de-inked. Nevertheless, the use of highly crosslinked cellulose acetate propionate (CAP) should be questioned due to its resistance against de-inking. Currently, the NC binder accounts for approximately 80% of the market for surface printing ink in Europe, the Middle East, and Africa (EMEA), and 55% of the packaging material placed in German yellow bins is printed [91]. The de-inking process might contribute positively to improving the quality of the post-consumer LDPE recyclate (e.g., 310 film fraction).

For the other types of binders, mainly PU and PVB, different opinions exist. The PU binder is reported to have a minor impact on the recyclate’s quality due to its higher heat stability [90]. However, it was also reported that other binder systems, including PU, PVB, acrylate, and CAP, tend to cause at least partial degradation under the extrusion temperature without gas forming [29]. The degradation of products and the impact on the recyclate’s quality is not clear [92]. Nevertheless, in a recycling plant, LDPE can tolerate a maximum of 3 wt.% rest humidity before extrusion. For a polycondensate polymer (e.g., PET, PU), the rest humidity is required to be reduced to 30–50 ppm, which means that a heat-stable PU binder can be degraded in an industrial recycling process [93,94,95]. In the context of a de-inking process, PU and PVB require higher temperatures (70 °C), which increases the energy consumption in the recycling process. Due to the low market share of PU and PVB ink for surface printing ink, it might not be economical to design the de-inking process for these more resistant binders.

UV-crosslinked acrylate is mainly used on rigid plastic packaging (e.g., the labels of PET bottles and in-mold labeling). PP-rigid packaging could benefit from recycling in a hot-washing process with only NaOH. This could remove the print color and improve the recyclate quality. Most of the PP stream is white mass colored with pressure-sensitive labels attached to it (which might be removed in the recycling process) or in-mold labels that can be de-inked.

Ink removal may be essential for the circularity of printed plastics [57]. In conclusion, the “design-for-recycling” principle for plastic printing ink should focus on the following aspects:The printing ink should retain its performance under chemical, heat, and mechanical stress (scratches and abrasion), ensuring resistance for application;b. The printing ink should preferably be de-inkable to yield light-colored recyclates without pigments.

### 4.2. Extension of the De-Inking Mechanism

The de-inking mechanism was reported by [59] and introduced in detail in the review part of this article. In light of the results of this study, it is necessary to consider an extension of this de-inking mechanism since this study is the first with samples of a known ink composition. The bonding mechanism of the binder systems can be categorized as (1) only with Van der Waals forces and hydrogen bonds due to the polar groups in the binder system and the activated plastic film surface after corona treatment; (2) binders with a reactive crosslinked system, where covalent bonds also exist to enhance the bonding between the binder and the plastic film surface. Among all the binder systems, the de-inking effect under condition A is, in general, low. The highest de-inking effect under condition A was achieved with detergent III. The enhanced de-inking effect under condition B was attributed to the increase in the pH values. Under alkaline conditions, the polymer surface carries a negative charge, facilitating the adsorption of cationic surfactants [80]. The process of de-inking in a non-crosslinked binder system is shown in Figure 11.

Additionally, only a surfactant cannot break the covalent bond between the binder and the plastic film [58]. For PVB, UV-crosslinked acrylate, and crosslinked nitrocellulose, chemical reactions such as hydrolysis can break the covalent bonds between the printing ink binder and plastic surface. Surfactants can hinder the de-inking effect by forming a protective layer on the printing ink surface when the HLB value is too low. This process is shown in Figure 12.

The de-inking mechanism can be revised as follows:Adsorption of the surfactant (ion pairing for cationic surfactant and hydrophobic bonding for nonionic surfactant);Breaking the covalent bonds via chemical reaction and hydrogen bonds via solubilization;Micelle solubilization (for non-crosslinked binder) and mechanical peel off (for crosslinked binder), to detach the printing ink from the plastic surface;Stabilization of the ink surface.

### 4.3. Inclusion of De-Inking Unit in Recycling Plants

Currently, some advanced recycling plants already apply hot washing at 70–80 °C with NaOH [18,96]. Hot washing is also reported as a possible solution for the degradation of plastic, especially for polyolefins [72,97,98]. Since a high cleaning effect was already reached in this study under condition B (55 °C and 1 wt.% NaOH), and washing with a detergent is also reported to be able to reduce the volatile organic compound (VOC) content in the recyclate from post-consumer plastic waste [99], incorporating a surfactant washing process may improve the quality of flexible polyolefin recyclates from post-consumer sources with lower energy consumption. The integration of the de-inking process into existing recycling plants, either by directly adding surfactants to the current washing unit or installing an additional hot-washing unit, should be further explored.

A critical concern in implementing detergent washing in industrial-scale recycling plants is the potential ecotoxicological impact. Cationic surfactants with tertiary amine groups (detergent II and IV) exhibit greater toxicity than nonionic surfactants (detergent III). Nevertheless, in industrial recycling plants, the process water used for cleaning plastic flakes is typically cleaned and reused. The concentration of surfactants in the wastewater of plastic recycling plants may be further diminished after treatment before being discharged. However, it is also crucial to conduct additional research to assess whether incorporating the de-inking process presents new challenges for the wastewater treatment process in plastic recycling plants.

## 5. Conclusions

In this study, the effectiveness of various de-inking methods in removing printing ink from flexible plastic packaging was investigated. The results provide a reference for developing de-inkable printing ink for plastic packaging and new detergent formulations for de-inking applications. It was observed that the type of pigment used in printing ink has a minimal impact on the de-inking efficiency, with the exception of carbon black (PBk 7). Among the binders tested, the single-layered NC binder demonstrated high de-inking efficiency when moderately heated (55 °C) and combined with 1 wt.% NaOH. It was even partially de-inkable under cold-washing conditions. Overprint varnishes can also be removed through de-inking processes if they have only moderate crosslinking. If crosslinked overprint varnishes are required for packaging design purposes, the development of a de-inkable system is recommended..

The optimal surfactants for de-inking should possess a hydrophilic–lipophilic balance (HLB) value ranging between 7 and 9. To advance surfactant development, it is essential to prioritize high ink detachment efficiency at approximately 40 °C, moderate HLB values (7–9), low toxicity, and biodegradability. Furthermore, investigating the best combination of various surfactants to enhance the de-inking performance is encouraged. Sodium hydroxide (NaOH) is a critical component in the de-inking process, as it imparts a negative charge to the binder surface, facilitating surfactant adsorption. Regarding de-inking mechanisms, detergent washing is the primary mechanism for non-crosslinked binder systems, while alkaline hydrolysis is the predominant mechanism for crosslinked systems.

However, this study has several limitations: (a) experiments were conducted on a laboratory scale, without considering the full recycling process, including friction washing, centrifugation, and rinsing; (b) there is no straightforward method for quantitatively measuring the bonding force between printing ink and plastic or the detachment force of surfactant micelles; (c) the potential interactions between printing inks of different compositions have not been explored, which could be an issue when scaling up the process.

## Figures and Tables

**Figure 1 polymers-15-02220-f001:**
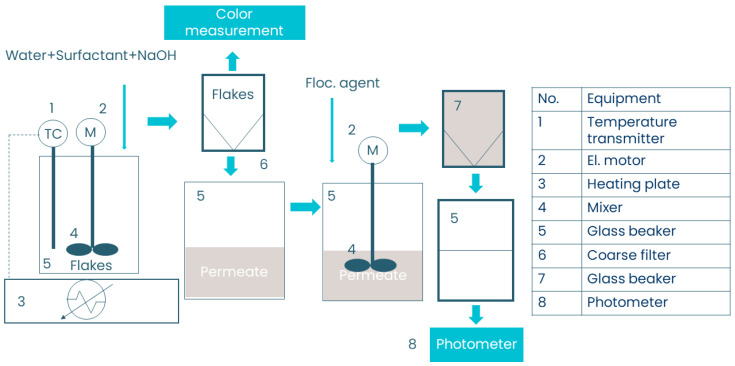
The experimental procedure and equipment.

**Figure 2 polymers-15-02220-f002:**
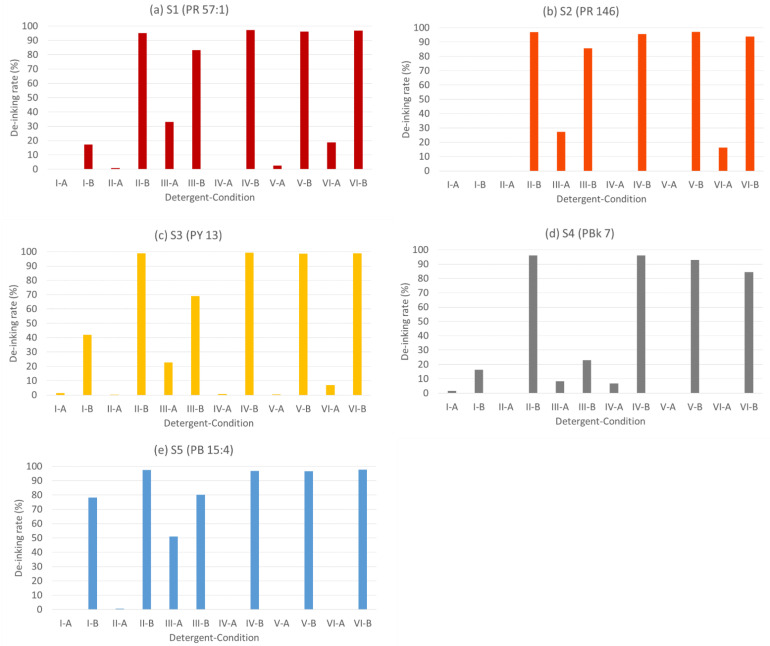
Influence of different pigments with different cleaning agents under conditions A and B (the colors of the bars represent the colors of the pigments) (**a**) Sample 1 with pigment PR 57:1,(**b**) Sample 2 with pigment PR 146, (**c**) Sample 3 with pigment PY 13, (**d**) Sample 4 with pigment PBk 7, (**e**) Sample 5 with pigment PB 15:4.

**Figure 3 polymers-15-02220-f003:**
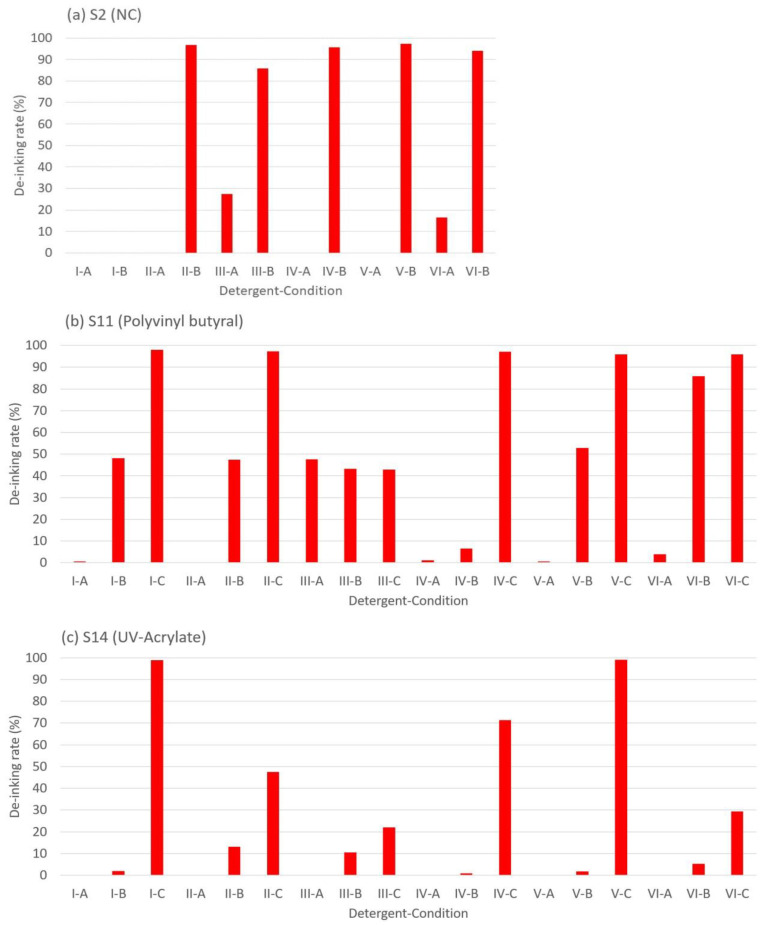
De-inkability of different binder systems: (**a**) NC binder, (**b**) PVB binder, (**c**) UV-crosslinked acrylate, without OPV, under conditions A, B, and C.

**Figure 4 polymers-15-02220-f004:**
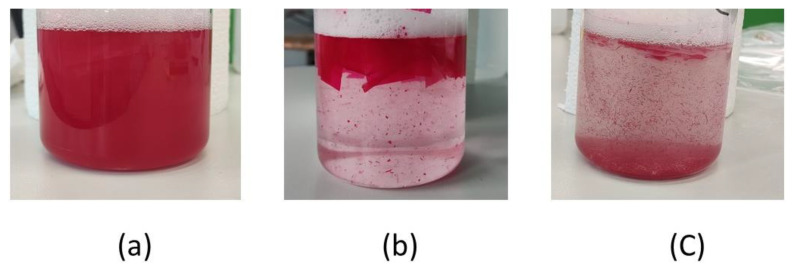
De-inking by solubilization and “peel off”: (**a**) NC binder, (**b**) PVB binder, (**c**) UV-crosslinked acrylate.

**Figure 5 polymers-15-02220-f005:**
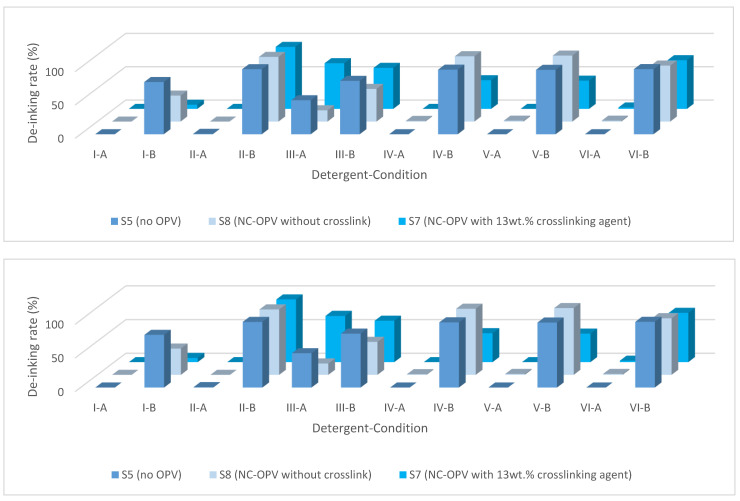
The influence of the crosslinking agent content on the de-inking rates under conditions A and B.

**Figure 6 polymers-15-02220-f006:**
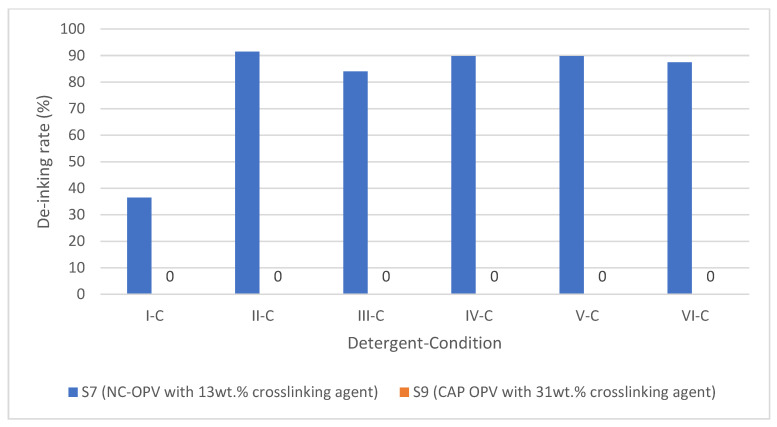
De-inking rates of samples with crosslinked OPV under condition C.

**Figure 7 polymers-15-02220-f007:**
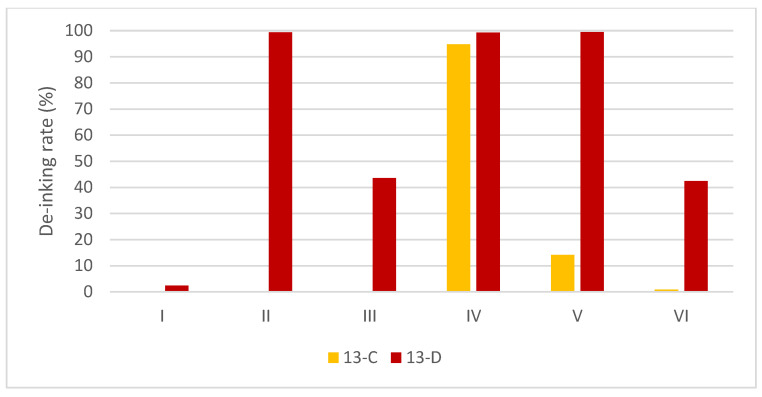
The influence of OPV of PU binder system on de-inking effects.

**Figure 8 polymers-15-02220-f008:**
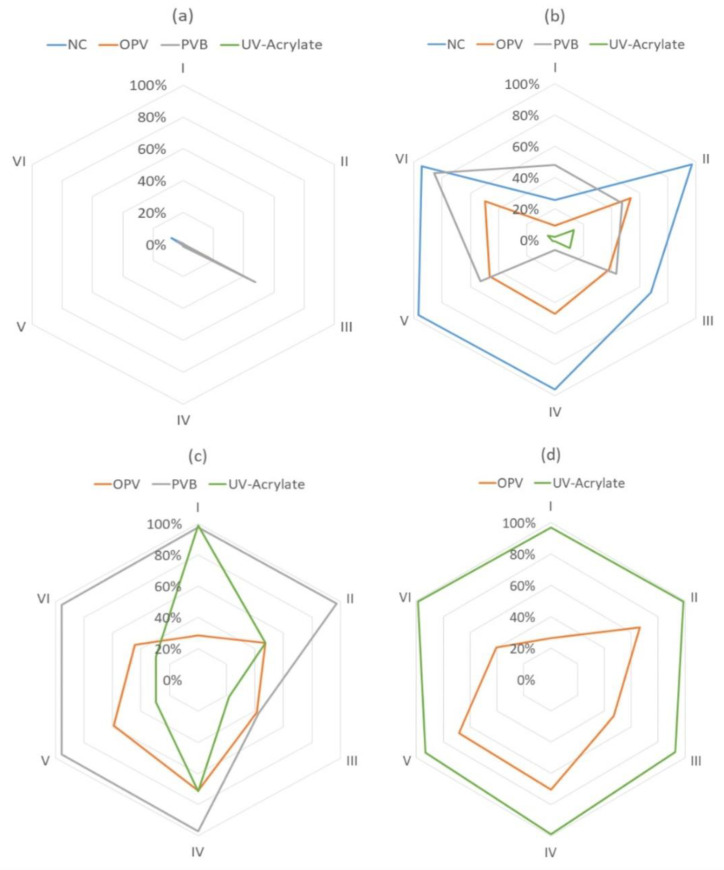
Total performance scores of the detergents tested under different conditions: (**a**) condition A, (**b**) condition B, (**c**) condition C, (**d**) condition D.

**Figure 9 polymers-15-02220-f009:**
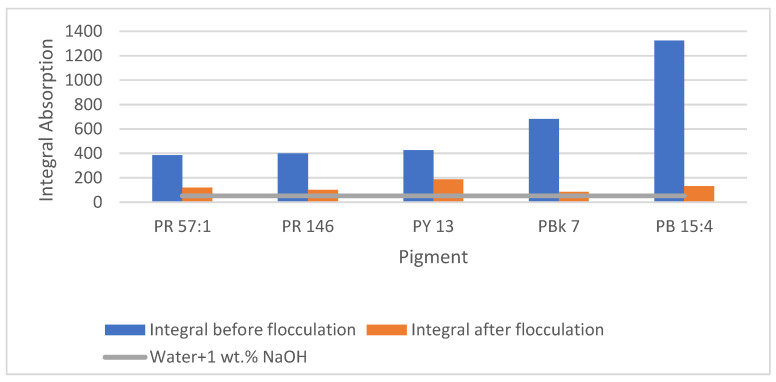
Changes in the light absorption of different pigments before and after flocculation.

**Figure 10 polymers-15-02220-f010:**
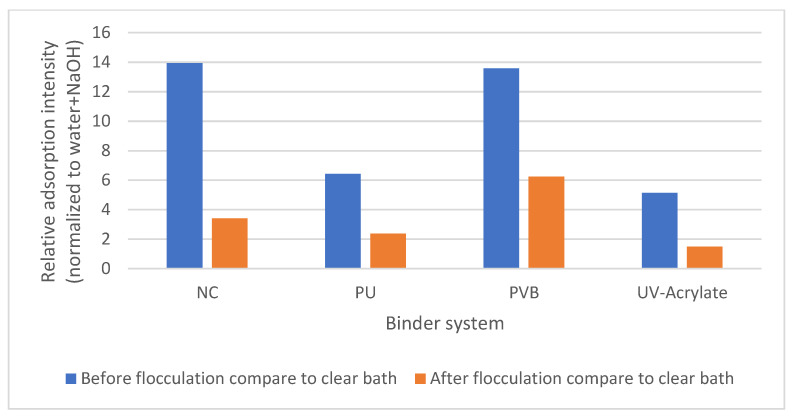
Relative absorption of the de-inking bath before and after flocculation for different binders.

**Figure 11 polymers-15-02220-f011:**
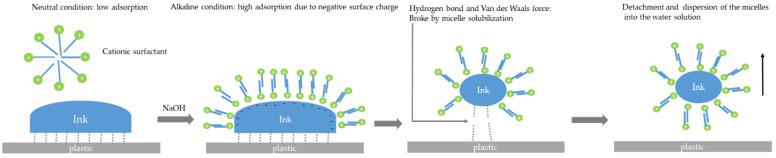
De-inking by detergent washing for non-crosslinked binder system.

**Figure 12 polymers-15-02220-f012:**
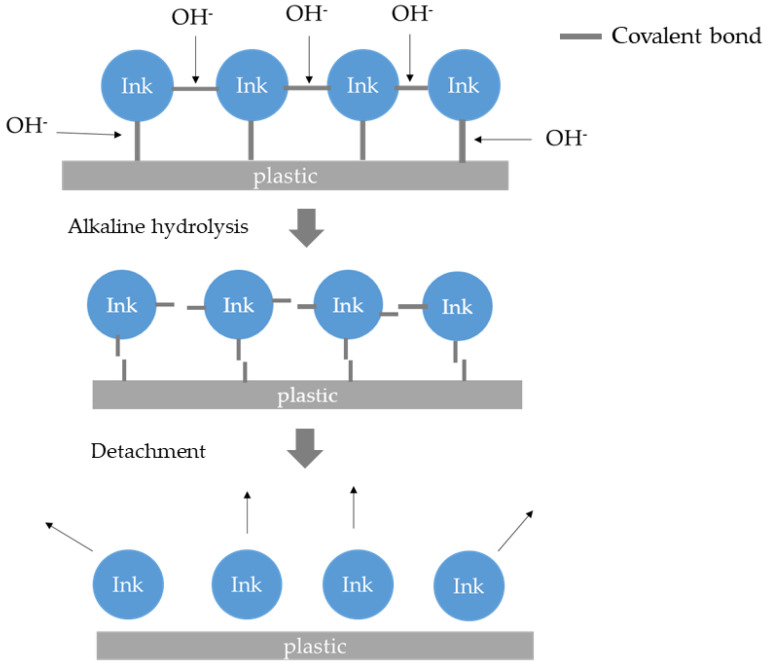
De-inking by chemical reaction (hydrolysis) for crosslinked binder systems.

**Table 1 polymers-15-02220-t001:** Typical composition of printing ink for plastic printing surface.

Plastic Printing Ink Component	Common Chemical Composition	Typical Content (wt.%)
Solvents	Organic solvents (e.g., alcohol, esters, ketones, etc.) for solvent-based ink Water with alcohols for water-based ink Liquid but reactive substances for UV and EB inks	50–70
Binders	Nitrocellulose (NC), polyurethane (PU), polyvinyl butyral (PVB), acrylate, cellulose acetate propionate (CAP), and mixtures thereof	20–30
Pigments	Organic and inorganic substances, named mainly by their color index (e.g., Pbk7)	6–30
Additives	Wax, surfactants, crosslinkers, dispersants, etc.	1–10

**Table 2 polymers-15-02220-t002:** An overview of previous de-inking studies.

De-Inked Sample	Surfactant	Surfactant Category	Result	Source
70% LDPE and 30% HDPE film, with water-based inks (color: black, white, purple, pink, and orange)	Sodium dodecyl sulfate (SDS)	Anionic	Partially de-inking above the CMC concentration at pH 8–9, increasing pH only contributed slight improvement	[64]
	Cetrimonium bromide (CTAB)	Cationic	Effective de-inking above CMC at pH 5–12	
	Hexadecylpyridinium chloride (CPC)	Cationic	Effective de-inking above CMC at pH 5–12	
	Nonylphenol polyethoxylated (NP(EO)10)	Nonionic	Almost complete de-inking at pH 10–12 above CMC	
	Dimethyl dodecyl amine oxide (DDAO)	Amphoteric	No relevance between pH and de-inking rate; effective de-inking effects were obtained at pH 5, 7, 11, and 12, above the CMC	
70% LDPE and 30% HDPE film, with water-based inks (color: black, white, purple, pink, and orange)	Ethoxylated alcohol (AEO5)	Nonionic	Effective ink removal above CMC at pH 10–12	[39]
	Ethoxylated amine (AMEO5)	Nonionic	Effective ink removal above CMC at pH 8–12	
70% LDPE and 30% HDPE film, with solvent-based inks (color: either yellow, pink, red, green, gold, black, or violet)	Sodium dodecyl sulfate (SDS)	anionic	No de-inking effect at pH 10–12	[65]
	Sodium dioctyl sulfosuccinate (SDOSS)	anionic	No de-inking effect at pH 10–12	
	Polyalkylene oxide modified polydimethylsiloxanes (molecular weight 600)	Nonionic	No de-inking effect at pH 10–12	
	Polyalkylene oxide modified polydimethylsiloxanes (molecular weight 1000)	Nonionic	No de-inking effect at pH 10–11, over 90% de-inking effect at pH 11.5 and 12	
	Dimethyl dodecylamine oxide (DDAO)	Amphoteric	No de-inking effect at pH 10–11, and slight de-inking effect at pH 11.5 and 12	
	Ethoxylated amine (AMEO5)	Nonionic	No de-inking effect at pH 10–11, and slight de-inking effect at pH 11.5 and 12	
	Cetrimonium bromide (CTAB)	Cationic	Complete de-inking at pH 11.5 and 12 with 1–2 h soaking	
Solvent-based ink with epoxy resin as a binder on HDPE bottles	DTAB	Cationic	Complete de-inking at 30 °C, 24 CMC, and 2 h retention time	[59]
	TTAB	Cationic	Complete de-inking at 30 °C, 8 CMC, and 2 h retention time	
	CTAB	Cationic	Complete de-inking at 30 °C, 4 CMC, and 2 h retention time	
Solvent-based ink with blue color printed on LDPE film	De-inking agent converted from PET named “GD-pyr-Br” and “GT-pyr-Br”	Cationic	GD-pyr-Br showed a de-inking effect at pH 12.5 and 13, while GT-pyr-Br was effective at pH 1–3	[66]
PE film with brown and green solvent-based printing inks; PP with red ink	De-inking agent from waste cooking oil	Cationic	70% de-inking effect after 15 h at pH 12 and 50 °C	[67]
Post-consumer LDPE film	W111+DA850	Nonionic	Effective (70–80%) de-inking in 15 min, at 40 °C and pH 12	[40]
Flexographic printed with different colors	Biosurfactant without known composition		Complete de-inking with 5% surfactant at 40 °C with brushing	[68]

**Table 3 polymers-15-02220-t003:** Test sample matrix (NC—nitrocellulose, PR—pigment red, PY—pigment yellow, PBk—pigment black, PB—pigment blue, CAP—cellulose acetate propionate); the chemical structures are listed in Appendix A.

Sample No.	Substrate	Binder	Pigment	OPV	Crosslinked Binder (y/n)	Crosslinked OPV (y/n)	Construction (Outside–Inside)
1	LDPE	NC	PR 57:1		n	n	Print–Substrate
2	LDPE	NC	PR 146		n	n	Print–Substrate
3	LDPE	NC	PY 13		n	n	Print–Substrate
4	LDPE	NC	PBk 7		n	n	Print–Substrate
5	LDPE	NC	PB 15:4		n	n	Print–Substrate
6	oPP	NC	PBk 7		n	n	Print–Substrate
7	LDPE	NC	PB 15:4	NC with 13 wt.% crosslinking agent	n	y	OPV–Print–Substrate
8	LDPE	NC	PB 15:4	NC	n	n	OPV–Print–Substrate
9	LDPE	NC	PR 146	CAP with 31 wt.% crosslinking agent	n	y	OPV–Print–Substrate
10	oPP	NC	PB 15:4	NC with 20 wt.% crosslinking agent	n	y	Print–OPV–Substrate
11	LDPE	PVB	PR 146	Release lacquer	n	n	Print–Substrate
12	LDPE	PU	PR 146		n	n	Print–Substrate
13	LDPE	PU	PR 146	PVB with 20 wt.% crosslinking agent	n	y	OPV–Print–Substrate
14	LDPE	Acrylate	PR 146	n	y	y	OPV–Print–Substrate

**Table 4 polymers-15-02220-t004:** Detergents used for the plastic de-inking experiment; the chemical structures and CAS numbers are listed in Appendix A.

No.	Surfactant	Source
I	No surfactant	[10]
II	Commercial cleaning agent (main component CTAC)	Siegwerk
III	W111+DA850	[40]
IV	CTAB	[59,65,69]
V	Triton X100	[70]
VI	Commercial cleaning agent for PET	[71]

**Table 5 polymers-15-02220-t005:** The test conditions.

Condition	NaOH (wt.%)	T (°C)	Surfactant (wt.%)	Retention Time (min)	Remarks
A	0	40	0.25	15	State of the art
B	1	55	0.25	15	“Middle point” condition
C	2	70	0.25	15	Hot washing condition
D	5	80	0.25	60	Only samples showed no de-inking effect under A–C

## Data Availability

The data presented in this study are available on request from the corresponding author.

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
