# Peer review of "Screening the Impact of Surfactants and Reaction Conditions on the De-Inkability of Different Printing Ink Systems for Plastic Packaging"

_polymers, 2023, doi:10.3390/polym15092220_

Round 1

Reviewer 1 Report

What does "7" in the Eq. 3 mean? Where does it come from?

Authors should change the number's separator from "," to "."

Moderate English changes are required.

Author Response

Dear Reviewer,

We would like to express our sincere gratitude for your valuable feedback and the time for reviewing our manuscript. These comments have played a significant role in enhancing the quality and presentation of the work.

Each comment and suggestion from the reviewers has been carefully considered and addressed in detail. In the attached document, a table outlining responses to each comment, along with the corresponding revisions made to the manuscript, can be found.

It is believed that these revisions have substantially improved the manuscript, and it is hoped that the updated version is now suitable for publication. Once again, thank you for the insightful comments and the efforts made in assisting with the improvement of this work.

Sincerely,

Jinyang Guo and all the co-authors

Reviewer 2 Report

The paper studied the Impact of Surfactants and Reaction Conditions on the De-inkability of Different Printing Ink Systems for Plastic Packaging. In general it is a good paper and the study is interesting. However, there some a few comments to improve this paper. 

1. The mechanism should be deeper, such as Section 4.2.

2. The figures are just the simple experimental results, and more description or analysis is better to be added in the figures.

Author Response

Dear Reviewer,

We would like to express our sincere gratitude for your valuable feedback and your time dedicated to reviewing our manuscript. These comments have played a significant role in enhancing the quality and presentation of the work.

Each comment and suggestion from the reviewers has been carefully considered and addressed in detail. In the attached document, a table outlining responses to each comment, along with the corresponding revisions made to the manuscript, can be found.

It is believed that these revisions have substantially improved the manuscript, and it is hoped that the updated version is now suitable for publication. Once again, thank you for the insightful comments and the efforts made in assisting with the improvement of this work.

Sincerely,

Reviewer 3 Report

The draft presented by the authors is interesting and deals with an important issue concerning the anthropic impact and undoubtedly deserving its publication. Nonetheless, the work needs to be improved, as it is very complex (due to the topics covered, the reagents and samples involved) and risks being difficult to read. In addition, it contains various inaccuracies and oversights, which I have highlighted in yellow trying to help the authors in the correction phase.  

Examples:

- temperatures often reported as "x°C", also appear in the text as "x °C" or "x°C",

- the decimal point is sometimes replaced by ",",

- “xwt.%”, sometimes “x wt. %” or “x wt.%”, etc…

Results

The term alkaline is chemically more appropriate of alkalic, which is more geological.

Line 369: “Figure 3(a)-(c). De-inkability of different binder (a)…, (b)… and (c) systems without OPV under condition A and B” ((a)…, (b)… and (c) similar to description of fig 4!)

And C, is missed?

Discussion

Line 548 What is 2k?

The point “4.4 Ecological Aspects” must be revised, it is not clear.

Line 551, “The increased de-inking effect measured under condition B can be concluded as the increased pH changes the charge on the surface of the polar binder molecules and favors the adsorption of the cationic surfactant [80].” Is not clear.

Conclusions

Authors affirm: “The main limitation of the current studies is: …“and then they report four points (a), b), c) and d))!

Technically, in point (b) the lack of literature data is not a limitation of the present work.

On line 542 (as well as in other parts of the text) it says "Chapter 0" or (Chap. x.x.x), but I don't understand if it is citing a book (a thesis!?!) or is it a paragraph of this draft. However, I suggest you substitute "paragraph" instead of "chapter" in your text.

In my opinion, paragraph has not been developed well and appears confusing.

Furthermore:

In the figure captions often, the dot is missed between “Figure n°” and next text.

Table 2 is the only to have a dot in the end of text.

The units (%) should be into the label of graph, not into the axis numeration for figure 2, 3, 5, 6, 7 and 10. The label on x axis for graph of fig. 10 is really complex, please check it.

Figure 9, the unit is missed on x axis.

In the graph of fig. 6 since the S9 sample does not appear, it is advisable to add a line or points to represent the sample at 0%.

Otherwise at first glance it seems like an oversight.

A question:

If there is no literature data on the chemical bonds between ink and packaging (as authors affirm at line 617), would it not have been better to investigate this aspect before trying different deinking processes blindly? This could represent a limit for the paper!

In conclusion, the authors must heavily revise the work to make it easier to read.

Author Response

Dear Reviewer,

We would like to express our sincere gratitude for your valuable feedback and your time dedicated to reviewing our manuscript. These comments have played a significant role in enhancing the quality and presentation of the work.

Each comment and suggestion from the reviewers has been carefully considered and addressed in detail. In the attached document, a table outlining responses to each comment, along with the corresponding revisions made to the manuscript, can be found.

It is believed that these revisions have substantially improved the manuscript, and it is hoped that the updated version is now suitable for publication. Once again, thank you for the insightful comments and the efforts made in assisting with the improvement of this work.

Sincerely,

Jinyang Guo and co-authors

Round 2

Reviewer 3 Report

The article has been revised by the authors according to the suggestions provided, so it seems appropriate to publish it.